# Self-Declared Roma Ethnicity and Health Insurance Expenditures: A Nationwide Cross-Sectional Investigation at the General Medical Practice Level in Hungary

**DOI:** 10.3390/ijerph17238998

**Published:** 2020-12-03

**Authors:** Feras Kasabji, Alaa Alrajo, Ferenc Vincze, László Kőrösi, Róza Ádány, János Sándor

**Affiliations:** 1Department of Public Health and Epidemiology, Faculty of Medicine, University of Debrecen, H-4012 Debrecen, Hungary; firas199141@gmail.com (F.K.); alaarajo@outlook.com (A.A.); vincze.ferenc@med.unideb.hu (F.V.); adany.roza@med.unideb.hu (R.Á.); 2Doctoral School of Health Sciences, University of Debrecen, H-4012 Debrecen, Hungary; 3Department of Financing, National Health Insurance Fund, H-1139 Budapest, Hungary; korosi.l@neak.gov.hu

**Keywords:** inequality, healthcare financing, general medical practice, health policy, self-reported Roma ethnicity

## Abstract

The inevitable rising costs of health care and the accompanying risk of increasing inequalities raise concerns. In order to make tailored policies and interventions that can reduce this risk, it is necessary to investigate whether vulnerable groups (such as Roma, the largest ethnic minority in Europe) are being left out of access to medical advances. **Objectives:** The study aimed to describe the association between general medical practice (GMP) level of average per capita expenditure of the National Health Insurance Fund (NHIF), and the proportion of Roma people receiving GMP in Hungary, controlled for other socioeconomic and structural factors. **Methods:** A cross-sectional study that included all GMPs providing care for adults in Hungary (*N* = 4818) was conducted for the period 2012–2016. GMP specific data on health expenditures and structural indicators (GMP list size, providing care for adults only or children also, type and geographical location of settlement, age of GP, vacancy) for secondary analysis were obtained from the NHIF. Data for the socioeconomic variables were from the last census. Age and sex standardized specific socioeconomic status indicators (standardized relative education, srEDU; standardized relative employment, srEMP; relative housing density, rHD; relative Roma proportion based on self-reported data, rRP) and average per capita health expenditure (standardized relative health expenditure, srEXP) were computed. Multivariate linear regression model was applied to evaluate the relationship of socioeconomic and structural indicators with srEXP. **Results:** The srEDU had significant positive (b = 0.199, 95% CI: 0.128; 0.271) and the srEMP had significant negative (b = −0.282, 95% CI: −0.359; −0.204) effect on srEXP. GP age > 65 (b = −0.026, 95% CI: −0.036; −0.016), list size <800 (b = −0.043, 95% CI: −0.066; −0.020) and 800–1200 (b = −0.018, 95% CI: −0.031; −0.004]), had significant negative association with srEXP, and GMP providing adults only (b = 0.016, 95% CI: 0.001;0.032) had a positive effect. There was also significant expenditure variability across counties. However, rRP proved not to be a significant influencing factor (b = 0.002, 95% CI: −0.001; 0.005). **Conclusion:** As was expected, lower education, employment, and small practice size were associated with lower NHIF expenditures in Hungary, while the share of self-reported Roma did not significantly affect health expenditures according to our GMP level study. These findings do not suggest the necessity for Roma specific indicators elaborating health policy to control for the risk of widening inequalities imposed by rising health expenses.

## 1. Introduction

Healthcare expenditure has risen dramatically during the last decades in the countries belonging to the Organization for Economic Co-operation and Development, where it accounts for 8.8% of the gross domestic product [1] and is expected to reach 14% by 2060 [2]. The growth is mainly driven by demands for restructuring elicited by ageing and rising level of chronic diseases, in addition to technological advances in health care [3,4]. As new technologies, facilities, and increased spending improve overall health outcomes, they also impose a risk of raising inequalities among social groups [5,6]. This trend is a serious concern for health policy makers.

Spontaneous market-based processes without strategic planning and purposeful execution of interventions cannot control for risks of inequality [7,8]. Effective public policy is required to counterbalance the inequality generated by technological development, incorporating community needs in planning, training programs, material aids, and grants for long term sustainability [9,10]. At present, there are striking inequalities in health spending in many respects, including individual, social, and healthcare system related determinants [11,12]. These factors can be the potential targets of interventions aimed at diminishing inequalities in spending.

Obviously, the demographic (age and sex, address [13,14,15]) and socioeconomic status (e.g., education [16,17,18], employment [19,20,21,22]) of clients influence expenditure [23]. The socioeconomic status of patients has a profound impact on the disease spectrum, and consequently on the needs for and the availability of high-quality care. The application of this well-established knowledge in financing primary medical care has a history of some decades [24,25,26,27,28]. General experience is that health spending shows huge geographical variation which is attributable mainly to socio-demographic factors. Moreover, the spatial variability of the health technology used and the organizational characteristics of institutions contribute considerably. [29]. Study of geographical resource allocation also has long-term history [30,31].

The most important health service in achieving equality is undoubtedly primary healthcare [32] for it has the utmost potential in influencing the levels of health inequalities. This comes from the fact that it covers the entire population [33]. It is the first contact with patients seeking care; moreover primary health care is stable and continuous and patients have many sessions with a known general practitioner (GP) from which they are never discharged [34]. Furthermore, primary healthcare allows for a cooperative relation between the GP and his patients allowing the GP to become an expert in their demographical and psychological constitution and their socioeconomic status, according to the declaration of Alma-Ata [35]. Moreover, GP interventions reduce the disadvantage some social groups have in access to higher tech medical treatments. The effect of the size of a general medical practice (GMP) on health outcome and spending was inconclusive in international studies [36,37,38]. Additionally, the age, gender and vacancy of a GP seems to have only limited effect [37,39,40].

Similarly, race and ethnicity’s influence on spending is well characterized [41,42,43,44], therefore the Jarman score adjusts for ethnicity. Contrarily, the Carstairs model and the Carr-Hill resource allocation formula do not input data on ethnic composition of the population provided [45,46]. Moreover, Roma ethnicity is not taken into consideration in routine heath statistics and resource allocation processes.

The Roma are the largest ethnic minority group in Europe [47]. Health indicators data is scarcely available, but generally they have higher rates of certain diseases [48,49]. These observations allow us to consider Roma as a risk factor for health. Many studies have been conducted to understand the mechanism by which Roma ethnicity leads to misuse of health facilities [50,51] and health loss [47,49,52], but our knowledge is still insufficient. This limitation is reflected in the hardly detectable impact of Roma targeted health policies in European countries [10,53,54]. To conclude, the full effect of the Roma population on the use of different health care services and financing is still not sufficiently documented. Moreover, knowing if being Roma and aspects of deprivation are separable or not is needed in order to implement successful polices that tackle inequality and better achieve universal coverage, for example, by incorporating the proportion of Roma people in the population provided by a GMP into the financing system as an adjustment factor. Evidently the structure of primary health care and the population share and culture of the Roma people is highly variable across European countries, therefore required interventions will have country specificity.

This study investigated the relation between the self-declared Roma proportion in the population served by GMPs, and payments to health services by the National Health Insurance Fund (NHIF) aggregated to the GMP level, while controlling for certain socioeconomic status indicators of patients and structural characteristics of GMPs. It examines whether the proportion of Roma is positively associated with NHIF expenditures reflecting that the poor health of the Roma population requires adequate care, or whether this correlation is negative, demonstrating that Roma’s poor access to health care counterbalances their increased needs. The necessity of GMP level Roma-specific indicators in formulating inequality-reducing health policy was also evaluated.

## 2. Methods

### 2.1. Setting

A nationwide cross-sectional study was conducted that included all GMPs (*N* = 4818) in Hungary that provide care for adults at least 18 years old. GMP-specific data on adult health expenditures for the period 2012–2016 and the structural characteristics of GMPs for 2012 were obtained from the NHIF which is contracted with each GMP operating in Hungary. Subjects of this investigations were Hungarian GMPs. Data on socioeconomic status indicators were taken from the most recent census undertaken by the Hungarian Central Statistical Office in 2011.

### 2.2. Explanatory Parameters: Socioeconomic Status Indicators

Data to compute socioeconomic indicators were provided by the Hungarian Central Statistical Office for settlements (residential places).

The observed number of self-declared Roma in each settlement was related to the expected number calculated by the total number of Roma in the country (*N* = 315,583), the whole population of Hungary (*N* = 9,937,628), and by the settlement’s population. The resulting indicator was the settlement specific relative Roma proportion (rRP).

The level of education (the number of years of school attendance) and the employment ratio for settlements were standardized by age and sex and summarized. The settlement specific expected numbers were calculated by national reference years of school attendance (among above at least seven year olds) and employment ratios (among above at least 15 year olds) by demographic strata (Appendix A
Table A1 and Table A2). Using demographic data of these settlements, standardized relative education (srEDU) and standardized relative employment (srEMP) were computed for each settlement.

The number of occupants per one hundred rooms was calculated as crowding index for each settlement. This was divided by the country average resulting in settlement specific relative housing density (rHD). There were 10,771,119 registered rooms in the country.

Because a GMP might have clients from more than one settlement, the settlement specific socioeconomic status indicators were transformed into GMP specific indexes. The settlement specific indicators were weighted according to the number of GMP’s clients living in different settlements.

Next, the resulting variables of srEMP, srEDU, srEXP, sRP were normalized using the two-step Box-Cox method [55,56]. These weighted and normalized GMP specific socioeconomic status indicators were used in further analyses.

### 2.3. Explanatory Parameters: GMP Structural Indicators

Multiple indicators were created for each GMP, such as the number of insured people who were registered in each GMP (categories were defined as ≤800, 801–1200, 1201–1600, 1601–2000 and >2000), the type of settlement in which the GMP operated (categorized as rural or urban), and the type of GMP by people served (adults only or adults and children). Since a typical Hungarian GMP is managed by one GP, GMPs with GP vacancies, with GPs < 65 years old, and with GPs ≥ 65 years old could be distinguished. GMPs were also categorized according to geographical location by county (Baranya, Bács-Kiskun, Békés, Borsod-Abaúj-Zemplén, Csongrád, Fejér, Gyor-Moson-Sopron, Hajdú-Bihar, Heves, Komárom-Esztergom, Nógrád, Pest, Somogy, Szabolcs-Szatmár-Bereg, Jász-Nagykun-Szolnok, Tolna, Vas, Veszprém, Zala and the capital Budapest.)

### 2.4. Outcome Variable

Health payments from the NHIF are basically divided between capitation fees and performance-based fees in Hungary. GMPs are financed by capitation fees, independent from services delivered by any provider at any level of healthcare. This takes into consideration only the number and demographic structure of clients belong to the GMP. Obviously, the average of the per capita age- and sex-standardized financing for GMPs’ clients is constant across the country, and not influenced by used health services.

Outpatient and inpatient secondary care, dental care, dialysis, imaging, transportation of patient, home nursing, and hospices are financed by performance, reimbursed by the NHIF according to the provided services. Medicines and medical devices, however, are financed by a co-payment system, sharing the costs between the NHIF and patients. Therefore, the variability of GMP level average financing by NHIF for GMPs’ clients is generated by these two types of financing [57].

The total of performance based reimbursements and co-payments for medicines and medical devices registered by the NHIF for adults belong to a GMP for primary medical service provision for the 5 years of the investigation, without per capita financing of primary medical care, were aggregated for GMPs. Expected 5-year expenditures and co-payments for medicines and medical devices were also calculated for each GMP by the 5-year per capita age and sex specific national reference payments (Appendix A
Table A3). The calculated ratio of the observed and expected expenditures resulted in the GMP specific standardized relative expenditure (srEXP).

### 2.5. Statistical Analysis

Per capita expenditures in a year for GMP categories, as the total of performance based reimbursements and co-payments for medicines and medical devices divided by the person-years of clients, were described by means (±SD). Uneven distribution of srEXP by socioeconomic characteristics of patients and structural characteristics of GMPs were tested with the Pearson correlation and one-way ANOVA.

A mixed two-level multivariate linear regression model (a) was used to investigate the influence of the GMP-specific socioeconomic indicators (rRP, srEDU, srEMP, rHD) and structural indicators (GP age and vacancy, type of settlement, GMP type, size measured by number of clients, and geographical location measured by county), taking into account the clustering effect of the counties. Linear regression coefficients (b) were used to describe the associations between explanatory variables and outcomes with corresponding 95% confidence intervals (95% CI). Goodness of fit was evaluated using the adjusted R^2^. Distribution of residuals, multicollinearity and heteroskedasticity issues were also investigated.

Three models were used to investigate the relationship between rRP and srEXP. A bivariate linear regression analysis (Model A), a multivariate model controlling for GMP structural characteristics (Model B), and Model B complemented with socioeconomic status indicators other than rRP (Model C) were implemented.

The standardized linear regression coefficients (β) with 95% CIs for Model C were calculated to determine the relative effect size of each independent variable.

SPSS version 20 (IBM Corporation, New York, NY, USA) was used for the data analysis.

## 3. Results

### 3.1. Descriptive Statistics

The total number of adults in the investigated GMPs was 7,506,059. The total expenditure was 873,797,515,655 HUF/year, with a national average per-capita expenditure of 116,412 HUF/year. Per-capita expenditures across demographic strata showed significant variation (Table A3). The average per-capita yearly GMP-specific expenditures also showed wide variability and were normally distributed (Figure 1). The mean (±SD) of the GMP-specific srEXP was 1 ± 0.15.

The GMP-specific srEDU, srEMP, rHD, and rRP had medians (IQRs) of 0.91 (0.1), 0.92 (0.22), 1.01 (0.20), and 0.37 (0.75) (Table 1). According to the analysis of Pearson’s correlation, rRP was negatively correlated with both srEDU (r = −0.55; *p* < 0.001) and srEMP (r = −0.71; *p* < 0.001). srEDU and srEMP were highly correlated with each other (r = 0.80; *p* < 0.001). The rHD was independent of each of the other socioeconomic indicators.

The number of GMPs with vacant GP positions was 274. GPs above the age of 65 were 26.70% of the total. Most of the GMPs were in urban areas (66.40%). Moreover, 69.30% of GMPs provided services to adults only. Most of the GMPs had list sizes of 1201–1600 (31.90%) and 1601–2000 (29.70%). In contrast, 19.20%, 4.00%, and 14.20% of GMPs provided care for more than 2000, less than 800, or 801–1200 insured patients, respectively (Table 2).

### 3.2. Regression Analysis

The two-level bivariate mixed linear regression model showed that rRP had a significant positive association with srEXP (b = 0.011, 95% CI: 0.008; 0.013). This relationship was confirmed by model B, which controlled for the confounding effects of GMP structural characteristics (b = 0.005, 95% CI: 0.002; 0.007). After complementing the model with other socioeconomic status indicators for the populations served, the rRP influence on srEXP proved to be nonsignificant (b = 0.002, 95% CI: −0.001; 0.005).

In model C (adjusted R^2^ = 0.147), srEDU (b = 0.199, 95% CI: 0.128; 0.271) had a positive association with expenditures, and srEMP (b = −0.282, 95% CI: −0.359; −0.204) had a negative association with srEXP. No significant relation was found between rHD and srEXP.

srEXP was significantly reduced in small GMPs that provide services for less than 800 (b = −0.043, 95% CI: −0.066; −0.020) and 800–1199 (b = −0.018, 95% CI: −0.031; −0.004) clients. Permanent GPs older than 65 years had a negative influence on spending (b = −0.026, 95% CI: −0.036; −0.016). GMPs that provided service to adults only had a significant positive association with srEXP (b = 0.016, 95% CI: 0.001; 0.032) compared to those providing service to both adults and children. Geographical location was also found to be a factor that significantly influenced expenditures (Table 3).

According to the standardized linear regression coefficients, srEMP (β = −0.219, 95% CI: −0.277; −0.159) had the strongest negative effect on spending, followed by being located in Győr-Moson-Sopron county (β = −0.140, 95% CI: −0.172; −0.108) and Veszprém county (β = −0.097, 95% CI: −0.128; −0.065). The strongest positive effect on spending came from being located in Baranya county (β = 0.159, 95% CI: 0.126;0.193) followed by srEDU (β = 0.13, 95% CI: 0.085;0.181). The nonsignificant effect of the Roma population proportion was associated with a very small standardized linear regression coefficient (β = −0.023, 95% CI: −0.017; 0.063). (Figure 2)

## 4. Discussion

### 4.1. Main Findings

Our results found a huge variability in GMP aggregated average per-capita NHIF expenditure among Hungarian GMPs. Similar results were found in another study [39]. The regression model we created controlled for the demographic composition of populations provided by GMPs, the structural indicators of GMPs, and the socioeconomic status of the population provided, which explained 14.7% of this variability.

The observed negative association of rRP with age and sex standardized health expenditure in univariate analysis, which is consistent with studies from Hungary [58,59] and Slovakia [52,60], seems to be independent of the GMP structural indicators. However, the Roma effect becomes insignificant after taking into consideration other socioeconomic indicators with well-known influence on health expenditures, such as level of education and employment [19,61,62]. The role of Roma ethnicity proved to be negligible compared to the significant role of GMP structural indicators which are proxy measures of the quality of primary care and of the socioeconomic status of clients indicated by their education and employment.

A positive association between a patient’s educational attainment and NHIF expenditures was observed in our investigation. This can be explained by the fact that people with higher levels of education are more attentive to their health and better understand their rights and the services provided to them by their GMPs [61,62]. Furthermore, they might have better connections with healthcare providers due to their higher social status, ultimately leading to higher usage of medical services, thus increasing NHIF expenditures. On the contrary, according to our multivariate regression analysis, there was a significant decrease in healthcare expenditure associated with employment. These results agree with studies that found unemployment to be associated with impaired overall health status [21,22] as well as an increase in utilization of healthcare facilities. Even though, generally, higher education leads to higher employment rates, these two variables were inversely related to NHIF expenditures. Time restrictions and the fear of losing one’s job among employed people lead to postponing the seeking for care. Contrary, unemployed people need more care because of their impaired health status, have more time to use health services, and many unemployed persons attempt to get a diagnosis which can result in disability pension [20,21]. The level of employment and education is remarkably low among Roma [59,63]. The opposite influences of employment and education on reimbursement counterbalance each other.

In Hungary, the typical GMP is operated by one GP and one nurse [57]. Therefore, a bigger list size as positive determinant of reimbursement does not correspond with the time given by the GP for services. Rather, the free choice of GPs in Hungary, which results in shifting patients from less to more intensive care, may be responsible for this finding. The negative influence of retirement, age of GP and the positive influence of more specialized (adults only) GMPs are in good concordance with the published observations [37,64,65].

Geographical location was also found to have strong impact on NHIF expenditures. These differences in spending could be attributed to the non-controlled confounding effects of county level specialties in healthcare services, or to social conditions and environmental circumstances. The same results were shown by international studies on the effect of geographical location on healthcare expenditure [12,66,67]. Because the county impact is remarkable, studies are needed to explore the details of the observed associations.

### 4.2. Strengths and Limitations

The most important strength of our study was that it covered the entire population of Hungary using census data, participation in which is compulsory, and including all GMPs in the country. Therefore, selection bias was effectively controlled.

Additionally, the statistical power of our model, which was high due to the nationwide design, was further increased by aggregating expenditures through a period of five years. Due to the achieved power, the lack of significance influence of rRP in population provided by GMPs is fairly convincing.

Interpretations of our study are determined by the missing data on the health status of persons belonging to a certain GMP. We could not control for the confounding effect of clients’ needs. Further investigations are required to control for this bias.

The cross-sectional nature of our study restricted the interpretation of the observed associations. However, the socioeconomic status and the GMP structural indicators applied do not change over a short time. The stability of explanatory variables diminishes the usual restrictions in interpretation of the cross-sectional observations. On the other hand, changes in socioeconomic status during the one to five years between the census and reimbursement data might reduce the validity of the reported associations from the regression models.

The ecological design, focusing on group level data, of our analysis limits the interpretation of the reported results. Because the subjects of the analyses were Hungarian GMPs, person level overinterpretation is not allowed. Our results are not about the determinants of NHIF expenditures on individual Roma, but about the role of the proportion of Roma in a GMP on the average per capita per year NHIF expenditure.

An important limitation of this study was that disease profile, as a direct and important determinant of health care use, was not among the controlled confounding factors. However, the confounding effect of disease profile was partly controlled, since adjusting factors such as age, sex, education, employment, and urbanization are strongly associated with disease occurrence. Therefore, a significant proportion of the confounding effect of disease profile was controlled by using the listed factors either in standardization or in multivariate regression modelling. Obviously, further studies are needed to determine the importance of the confounding effect of disease pattern.

Because the NHIF and the Census 2011 of the Hungarian Central Statistical Office apply standardized protocols for data collection, measurement bias was negligible in the study, in general. However, Roma were identified by self-declaration in Census 2011 which resulted in serious underreporting [68,69]. This can mitigate considerably the observed influence of the rRP on expenditures. However, the underestimation for the role of Roma proportion by investigating self-declared figures did not prevent the observation of significant associations with srEXP in univariate Model A, and in the model B not controlled for the srEDU and srEMP. The rRP association with srEXP disappeared with the involvement of srEDU and srEXP in regression modelling suggesting that the strong correlation of rRP with srEDU and srEMP is responsible for the disappearing influence of rRP when extending Model B to Model C. However, it must be admitted that our results have to be interpreted for self-declared Roma, not for the whole population of Hungarian Roma.

### 4.3. Implications

This ecological investigation suggests that the proportion of self-declared Roma in the population provided by a GMP has no association with NHIF reimbursements independent of education and employment, two strong and well-known socioeconomic determinants of health status. This underlines the uncertainties about the justification for Roma specific policies [70,71] instead of focusing primarily on socioeconomic factors regardless of ethnicity. The mechanisms by which Roma ethnicity is converted to loss of health should be explored more thoroughly in order to tailor interventions against causal factors among Roma. More systematic research should be implemented to provide the required evidence for health policy formulation. The descriptive statistics are convincing on the critical health status of Roma across countries, making this an urgent issue, since former interventions have hardly been effective.

At present, there is no base on which to build the Roma population’s share of the health financing system in Hungary, which takes into consideration area level deprivation by type of settlement. However, building on employment and education could improve the suitability of the financing system. A Scottish index of multiple deprivation and the Indices of Deprivation 2007 used in the National Health Service in the United Kingdom incorporate education and employment among domains to describe deprivation for geographically defined populations, but not race and ethnicity [70,71].

## 5. Conclusions

Our GMP level aggregated data based investigation showed very large inequality in NHIF expenditures across Hungarian counties. The higher level of education and unemployment had significant positive effect on NHIF expenditure, making these prime targets for intervention. The self-reported Roma population had a nonsignificant association with NHIF expenditures. These ecological observations do not suggest that Roma’s access to health care is restricted. These observations suggest that the poor access of Roma to health care is a weaker or less effective risk factor in reduced NHIF financing than their poor health status which determines increased needs.

## Figures and Tables

**Figure 1 ijerph-17-08998-f001:**
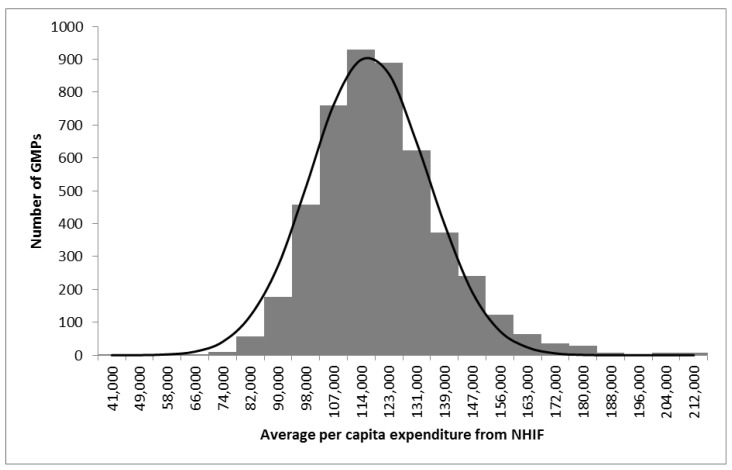
Distribution of average per-capita expenditure among the Hungarian general medical practices (GMPs) studied with the reference normal distribution curve.

**Figure 2 ijerph-17-08998-f002:**
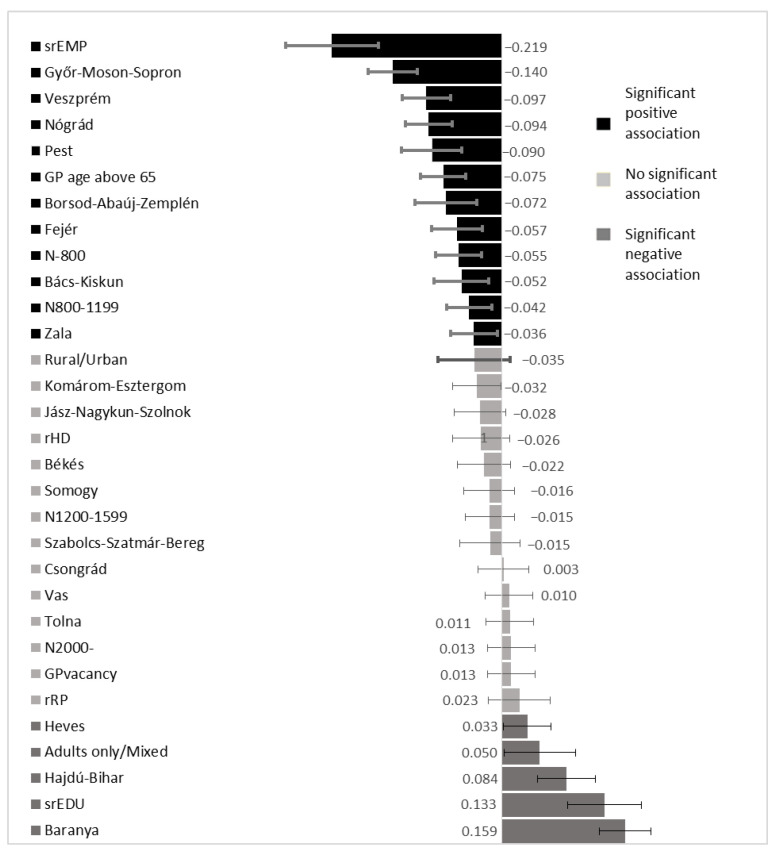
Strength of the association between socioeconomic factors, including Roma population proportion, and GMP-specific structural indicators with average per-capita GMP-specific expenditures of the National Health Insurance Fund based on the standardized linear regression coefficients from multivariate regression model. (srEMP: normalized standardized employment ratio, rHD: normalized relative housing density, srEDU: normalized relative education, rRP: normalized Roma proportion, N: number of patients at each GMP with reference N1600−1999 Budapest as the reference for counties).

**Table 1 ijerph-17-08998-t001:** Socioeconomic status indicators for the whole population and their distribution among general medical practices in Hungary.

Variable	Crude Indicator for the Whole Country	Median (IQR) for Relative GMP Specific Values
Roma proportion	3.10% (315,583/9,937,628)	0.54 (2.30)
Employment ratio	46.44% (3,942,723/8,489,969 *)	0.92 (0.22)
Housing density	1.08 (10,771,119 ******/9,937,628)	1.01 (0.20)
Years of education	10.38 (96,217,389/9264462 ***)	0.91 (0.1)

* population over 14 years old, ** number of rooms for a person, *** population over 7 years.

**Table 2 ijerph-17-08998-t002:** Per-capita expenditures (in Hungarian forint) of the National Health Insurance Fund by general medical practice (GMP) structural characteristics during the period 2012–2016 in Hungary.

GMP Characteristics	Categories	Number of GMPs (%)	Average per Capita Expenditure (±SD)	*p* Value *
GP (age and vacancy)	Vacant GMPs	273 (5.70%)	113,976 (±20,715)	0.023
	GPs younger than 65	3532 (73.30%)	116,759 (±20,943)	
	GPs older than 65	1289 (26.70%)	116,988 (±19,394)	
Type of settlement	Urban	3198 (66.40%)	118,042 (±19,091)	<0.001
	Rural	1620(33.60%)	114,408 (±22,951)	
GMP type	For adults only	3337 (69.30%)	117,982 (±19,043)	<0.001
	For adults and children	1481 (30.70%)	114,203 (±23,360)	
GMP size (number of patients)	≤800	193 (4.00%)	117,986 (±23,687)	<0.001
	801–1200	725 (15.20%)	119,346 (±22,918)	
	1201–1600	1540 (31.90%)	118,382 (±21,039)	
	1601–2000	1434 (29.70%)	116,347 (±19,585)	
	2000<	926 (19.20%)	112,735 (±17,671)	
County	Budapest	865 (18.00%)	117,989 (±18,068)	<0.001
	Baranya	209 (4.30%)	135,521 (±20,263)	
	Bács-Kiskun	256 (5.30%)	116,025 (±17,696)	
	Békés	187(3.90%)	122,870 (±21,309)	
	Borsod-Abaúj-Zemplén	372 (7.70%)	114,362 (±20,751)	
	Csongrád	204 (4.20%)	121,642 (±17,366)	
	Fejér	194 (4.00%)	111,226 (±19,894)	
	Győr-Moson-Sopron	203 (4.20%)	103,027 (±15,623)	
	Hajdú-Bihar	244 (5.10%)	124,405 (±19,657)	
	Heves	161 (3.30%)	124,883 (±21,479)	
	Komárom-Esztergom	144 (3.30%)	110,720 (±16,981)	
	Nógrád	109 (2.30%)	113,147 (±17,877)	
	Pest	481 (10.00%)	110,442 (±20,442)	
	Somogy	172(3.60%)	120,730 (±20,856)	
	Szabolcs-Szatmár-Bereg	266 (5.50%)	112,080 (±15,928)	
	Jász-Nagykun-Szolnok	194 (4.00%)	118,343 (±19,830)	
	Tolna	119 (2.50%)	121,861 (±17,970)	
	Vas	133 (2.80%)	117,535 (±33,926)	
	Veszprém	164 (3.40%)	109,797 (±16,885)	
	Zala	141 (2.90%)	116,221 (±19,183)	
Total	---	4818 (100.00%)	116,820 (±20,539)	---

* by one-way ANOVA.

**Table 3 ijerph-17-08998-t003:** Association between the proportion of Roma people in the population served by a GMP and standardized normalized average per-capita expenditures of the National Health Insurance Fund in Hungary estimated with linear regression models controlling for the socioeconomic status of patients and the structural characteristics of GMPs.

		Model A		Model B		Model C	
Variables		B (95% CI) *	*p* Value	B (95% CI) *	*p* Value	B (95% CI) *	*p* Value
Roma proportion	(Normalized)	0.011 (0.008; 0.013)	<0.001	0.005 (0.002; 0.007)	0.001	0.002 (−0.001; 0.005)	0.250
Type of settlement	Rural			−0.007 (−0.022; 0.007)	0.329	−0.011 (−0.026; 0.004)	0.140
	Urban			1 (reference)		1 (reference)	
GP position	GP permanent, ≥65 years old			−0.026 (−0.036; −0.016)	<0.001	−0.026 (−0.036; −0.016)	<0.001
	GP vacancy			0.010 (−0.010; 0.030)	0.330	0.008 (−0.012; 0.028)	0.410
	GP permanent, <65 years old			1 (reference)		1 (reference)	
GMP type	GMP for adults only			0.016 (0.001; 0.031)	0.038	0.016 (0.001; 0.032)	0.040
	GMP for children and adults			1 (reference)		1 (reference)	
List size	≤800			−0.038 (−0.061; −0.015)	<0.001	−0.043 (−0.066; −0.020)	<0.001
	801–1200			−0.012 (−0.025; 0.001)	0.074	−0.018 (−0.031; −0.004)	0.010
	1201–1600			−0.003 (−0.013; 0.007)	0.576	−0.005 (−0.015; 0.005)	0.350
	1601–2000			1 (reference)		1 (reference)	
	2000<			0.003 (−0.009; 0.015)	0.592	0.005 (−0.007; 0.017)	0.420
County	Baranya			0.136 (0.114; 0.159)	<0.001	0.120 (0.094;0.145)	<0.001
	Bács-Kiskun			−0.030 (−0.051; −0.010)	<0.001	−0.035 (−0.059; −0.011)	<0.001
	Békés			−0.002 (−0.024; 0.021)	0.883	−0.017 (−0.044; 0.010)	0.210
	Borsod-Abaúj-Zemplén			−0.021 (−0.040; −0.002)	0.027	−0.041 (−0.064; −0.018)	<0.001
	Budapest			1 (reference)		1 (reference)	
	Csongrád			0.012 (−0.01; 0.034)	0.271	0.002 (−0.023; 0.027)	0.880
	Fejér			−0.047 (−0.069; −0.024)	<0.001	−0.045 (−0.070; −0.019)	<0.001
	Győr-Moson-Sopron			−0.129 (−0.152; −0.107)	<0.001	−0.106 (−0.131; −0.082)	<0.001
	Hajdú-Bihar			0.080 (0.060; 0.101)	<0.001	0.058 (0.032; 0.084)	<0.001
	Heves			0.037 (0.012; 0.062)	<0.001	0.028 (0.002; 0.055)	0.040
	Jász-Nagykun-Szolnok			−0.016 (−0.039; 0.006)	0.158	−0.021 (−0.047; 0.004)	0.100
	Komárom-Esztergom			−0.056 (−0.081; −0.030)	<0.001	−0.028 (−0.056; 0)	0.050
	Nógrád			−0.079 (−0.108; −0.050)	<0.001	−0.096 (−0.127; −0.065)	<0.001
	Pest			−0.043 (−0.06; −0.026)	<0.001	−0.046 (−0.065; −0.026)	<0.001
	Somogy			0.002 (−0.022; 0.026)	0.848	−0.013 (−0.040; 0.014)	0.350
	Szabolcs-Szatmár-Bereg			0.010 (−0.011; 0.031)	0.359	−0.010 (−0.036; 0.016)	0.460
	Tolna			0.016 (−0.012; 0.044)	0.255	0.011 (−0.019; 0.041)	0.490
	Vas			−0.022 (−0.048; 0.005)	0.110	0.009 (−0.019; 0.037)	0.530
	Veszprém			−0.089 (−0.113; −0.065)	<0.001	−0.082 (−0.108; −0.055)	<0.001
	Zala			−0.047 (−0.072; −0.021)	<0.001	−0.032 (−0.059; −0.005)	0.020
Employment	(Normalized)					−0.282 (−0.359; −0.204)	<0.001
Housing density	(Normalized)					−0.034 (−0.082; 0.014)	0.160
Education	(Normalized)					0.199 (0.128; 0.271)	<0.001

* Linear regression Coefficient (B) and 95% confidence interval (95% CI).

## Data Availability

The datasets used and/or analyzed during the current study are available from the corresponding author on reasonable request.

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
