# Peer review of "Self-Declared Roma Ethnicity and Health Insurance Expenditures: A Nationwide Cross-Sectional Investigation at the General Medical Practice Level in Hungary"

_ijerph, 2020, doi:10.3390/ijerph17238998_

Round 1
Reviewer 1 Report
The paper deals with health insurance reimbursement for population of the Roma ethnicity. It is based on a nationwide cross-sectional investigation, but the data comes from the most recent national census in fact. It sounds little bit misleading. The explained variable is mentioned in the 2.4 subsection, but the explanatory variables are missing. They are mentioned for the first time in Table 1 as the regressors. Additionally, the explained variable is mentioned in a very vague way. What is its unit? Also, the units of the regressors and the whole description is missing before this table. It is quite important, because interpretation of the regression coefficients is the most crucial outcome of this type analysis.
In the 2.5 subchapter, all the methodological approaches are listed and described. As you carry out the regression analysis, it is required to perform also the test in order to avoid errors in the interpretation of its outcome. Have you done tests for multicollinearity and heteroscedasticity for instance? Have you checked whether the residual values possess normal distribution of probability?
In the fifth paragraph of the 4.2 subchapter, you mention that because of the ecological design of the study, the interpretation of the results is limited. What is meant by it? Or what do you mean by so-called "ecological design"? Another point is that "the person level overinterpretation is not allowed" according to you. It could be very disputable, because just right interpretation is the most important point of the whole analysis.
The text of the paper involves a mixture of the British English and the American English. It is needed to unify it into the particular version.
There are also some formal mistakes, but they do not lower quality of the paper significantly – for instance, missing spaces in front of the left brackets opening the references in the text for many times throughout the text. The second reference is not listed as it should be – only its name and year is mentioned.
Author Response
Dear Editor and Reviewer,
Thank you very much for the careful evaluation of our manuscript “Self-declared Roma ethnicity and health insurance reimbursement: a nationwide cross-sectional investigation at the general medical practice level in Hungary”. Each comment and suggestion has been considered. Answers and responses along with the modifications we made are the following (Reviewer’s comments and questions are in capitals):
Reviewer-1:
1.
THE PAPER DEALS WITH HEALTH INSURANCE REIMBURSEMENT FOR POPULATION OF THE ROMA ETHNICITY. IT IS BASED ON A NATIONWIDE CROSS-SECTIONAL INVESTIGATION, BUT THE DATA COMES FROM THE MOST RECENT NATIONAL CENSUS IN FACT. IT SOUNDS LITTLE BIT MISLEADING.
The cross-sectional nature if the investigation refers to the fact that the NHIF (National Health Insurance Fund) as well as the Hungarian Central Statistical Office collect data continuously ,recent data are associated by demographic structure a decade ago (2011) ,however as is well known socioeconomic status does not change promptly. To admit explicitly this validity issue a sentence has been inserted into the 4.2 Strengths and limitations section (paragraph 4):
“On the other hand, changes of socio-economic status during the 8 years between the census and reimbursement data reduce the validity of the reported associations from the regression models.”
2.
THE EXPLAINED VARIABLE IS MENTIONED IN THE 2.4 SUBSECTION, BUT THE EXPLANATORY VARIABLES ARE MISSING. THEY ARE MENTIONED FOR THE FIRST TIME IN TABLE 1 AS THE REGRESSORS.
Explanatory variables such as the socioeconomic variables and the GMP structural variables are mentioned in sections 2.2 and 2.3. To make it more obvious that these groups of parameters are explanatory parameters the subtitles has been modified:
Original:
2.2. Socioeconomic status indicators
2.3. GMP structural indicators
Corrected
2.2. Explanatory parameters: Socioeconomic status indicators
2.3. Explanatory parameters: GMP structural indicators
3.
ADDITIONALLY, THE EXPLAINED VARIABLE IS MENTIONED IN A VERY VAGUE WAY. WHAT IS ITS UNIT?
The vague explanation of the dependent variable in the 2.4 section has been replaced with a more detailed one, and a reference has been added as well.
Original:
Hungarian GMPs are financed basically by capitation fees from NHIF. It is independent of services delivered by any providers from any level of healthcare. Therefore, GMP financing by NHIF does not influence the per capita health spending variability.
The total of expenditures registered by the NHIF without the per capita financing for GMPs for primary medical service provision for the 5 years of the investigation among adults were aggregated for GMPs. Expected 5-year payment were also calculated for each GMP by the 5-year per capita age and sex specific national reference payments. (Appendix Table A3) The calculated ratio of the observed and expected per capita payments resulted the GMP specific standardized relative expenditure (srEXP).
Corrected:
Health reimbursement from the NHIF is basically divided by capitation fees and performance-based fees in Hungary. GMPs are financed by capitation fees independent of services delivered for their patients by any providers from any level of healthcare. It takes only into consideration the number and demographic structure of the clients belong to the GMP. Obviously, the average of the per capita age- and sex-standardized financing for clients belong to GMP is constant across the country, and not influenced by the used health services. The outpatient and inpatient secondary care, medicine and medical device reimbursement, dental care, dialysis, imaging, transportation of patient, home nursing, and hospice are financed by performance. Therefore, the variability of the GMP level average financing by NHIF for clients belong to a GMP does generated by the use of these services. [64].
The total of expenditures registered by the NHIF for adults belong to a GMP for primary medical service provision for the 5 years of the investigation, without the per capita financing of primary medical care, were aggregated for GMPs as average per capita per year expenditure. Expected 5-year payment were also calculated for each GMP by the 5-year per capita age and sex specific national reference payments. (Appendix Table A3) The calculated ratio of the observed and expected per capita payments resulted the GMP specific standardized relative expenditure (srEXP).
4.
ALSO, THE UNITS OF THE REGRESSORS AND THE WHOLE DESCRIPTION IS MISSING BEFORE THIS TABLE. IT IS QUITE IMPORTANT, BECAUSE INTERPRETATION OF THE REGRESSION COEFFICIENTS IS THE MOST CRUCIAL OUTCOME OF THIS TYPE ANALYSIS.
The missed information that the level of education was indicated by the years of school attendance, and the level of employment was indicated by the proportion of employed persons have been added to the second paragraph of the 2.2 section.
Original:
The level of education and the employment for settlements were standardized by age and sex. Number of years of school attendance and the number of employed persons were summarized for settlements.
Corrected:
The level of education (as the number of years of school attendance) and the employment ratio for settlements were standardized by age and sex and summarized for settlements.
5.
IN THE 2.5 SUBCHAPTER, ALL THE METHODOLOGICAL APPROACHES ARE LISTED AND DESCRIBED. AS YOU CARRY OUT THE REGRESSION ANALYSIS, IT IS REQUIRED TO PERFORM ALSO THE TEST IN ORDER TO AVOID ERRORS IN THE INTERPRETATION OF ITS OUTCOME. HAVE YOU DONE TESTS FOR MULTICOLLINEARITY AND HETEROSCEDASTICITY FOR INSTANCE? HAVE YOU CHECKED WHETHER THE RESIDUAL VALUES POSSESS NORMAL DISTRIBUTION OF PROBABILITY?
Tests for multicollinearity, heteroskedasticity and normality of residuals were carried out. There was no multicollinearity according to multicollinearity diagnostics in SPSS where no variable had a VIF (variance inflation factor) >10. As for heteroskedasticity, our data is homoscedastic according to SPSS plot analysis of residuals.
Our analysis also shows that although residuals according to Kolmogorov-Smirnov are not normally distributed, the shape of the distribution by and large still followed normal distribution (Figure 2). The statistically significant deviation from the normal distribution is attributable to the extreme statistic lower of the test due to the extreme size of our sample.
Relevant graphs and tables were added to the appendix section for more clarity on the subject. The Statistical analysis section has been completed:
Original:
Goodness of fit was evaluated using the adjusted R2.
Corrected:
Goodness of fit was evaluated using the adjusted R2. Distribution of residuals, multicollinearity and heteroskedasticity issues were also investigated.
6.
IN THE FIFTH PARAGRAPH OF THE 4.2 SUBCHAPTER, YOU MENTION THAT BECAUSE OF THE ECOLOGICAL DESIGN OF THE STUDY, THE INTERPRETATION OF THE RESULTS IS LIMITED. WHAT IS MEANT BY IT? OR WHAT DO YOU MEAN BY SO-CALLED "ECOLOGICAL DESIGN"? ANOTHER POINT IS THAT "THE PERSON LEVEL OVERINTERPRETATION IS NOT ALLOWED" ACCORDING TO YOU. IT COULD BE VERY DISPUTABLE, BECAUSE JUST RIGHT INTERPRETATION IS THE MOST IMPORTANT POINT OF THE WHOLE ANALYSIS.
Since our investigation encompasses the entirety of the Hungarian population and not individuals, it is considered an ecological study. Clarified meaning has been added to the relevant paragraph.
Original:
The ecological design of our analysis limits the interpretation of the reported results. Because the subjects of the analyses were the Hungarian GMPs, the person level overinterpretation is not allowed. Our results are not about the determinants of the Roma individuals’ NHIF health reimbursement.
Corrected:
The ecological design focused on group level data of our analysis limits the interpretation of the reported results. Because the subjects of the analyses were the Hungarian GMPs, the person level overinterpretation is not allowed. Our results are not about the determinants of the Roma individuals’ NHIF health reimbursement, but about the role of the proportion of Roma in a GMP on the average per capita per year NHIF reimbursement.
7.
THE TEXT OF THE PAPER INVOLVES A MIXTURE OF THE BRITISH ENGLISH AND THE AMERICAN ENGLISH. IT IS NEEDED TO UNIFY IT INTO THE PARTICULAR VERSION.
The paper’s chosen spelling was the American one. British spelled words were changed to American spelling as follows:
In the Author contributions and Data availability sections the word “analysed” was changed to “analyzed”.
In the Introduction (paragraph 5), the word “characterised” was changed to “characterized”.
8.
THERE ARE ALSO SOME FORMAL MISTAKES, BUT THEY DO NOT LOWER QUALITY OF THE PAPER SIGNIFICANTLY – FOR INSTANCE, MISSING SPACES IN FRONT OF THE LEFT BRACKETS OPENING THE REFERENCES IN THE TEXT FOR MANY TIMES THROUGHOUT THE TEXT.
All brackets throughout the paper were corrected according to this suggestion.
9.
THE SECOND REFERENCE IS NOT LISTED AS IT SHOULD BE – ONLY ITS NAME AND YEAR IS MENTIONED.
Fixed the second reference being incomplete as follows:
Original:
WHAT FUTURE FOR HEALTH SPENDING?; 2013.
Corrected:
What Future for Health Spending? OECD; 2013. OECD Economics Department Policy Notes, No. 19 June 2013.
Reviewer 2 Report
Please find some comments that may be useful below:
Abstract:
“Objectives: The study aimed to describe the association between general medical practice (GMP) level average per capita health care reimbursement and proportion of Roma people belonging to the GMP in Hungary, controlled for other socioeconomic and GMP structural factors (GMP list size, providing care for adults only or children also, type and geographical location of settlement, age of GP, and vacancy)”. This sentence seems to have combined methods with aims/objects.
In the results section, please focus on interpreting the main findings rather than providing the point estimates only. For example, what does it mean to have a b=0.199. Also, please provide CI and not p-values here.
Introduction:
This section is very well-drafted overall.
OECD- please provide the fullform; also please try to limit the abbreviations used
Methods:
There are way too many acronyms in this section. Please try and limit the use of acronyms.
Since expenditures cannot be less than 0, did the authors explore using other distributions like gamma?
What does ANOVA add when the authors already have estimates from adjusted regression models?
On what basis were the covariates to be adjusted for chosen?
Results:
Instead of just providing the beta and p-values, please rather focus on interpretation and provide CI instead of or along with p-values
Discussion and conclusions:
Well drafted! No further comments
Author Response
Dear Editor and Reviewer,
Thank you very much for the careful evaluation of our manuscript “Self-declared Roma ethnicity and health insurance reimbursement: a nationwide cross-sectional investigation at the general medical practice level in Hungary”. Each comment and suggestion has been considered. Answers and responses along with the modifications we made are the following (Reviewer’s comments and questions are in capitals):
Reviewer-2:
1.
“OBJECTIVES: THE STUDY AIMED TO DESCRIBE THE ASSOCIATION BETWEEN GENERAL MEDICAL PRACTICE (GMP) LEVEL AVERAGE PER CAPITA HEALTH CARE REIMBURSEMENT AND PROPORTION OF ROMA PEOPLE BELONGING TO THE GMP IN HUNGARY, CONTROLLED FOR OTHER SOCIOECONOMIC AND GMP STRUCTURAL FACTORS (GMP LIST SIZE, PROVIDING CARE FOR ADULTS ONLY OR CHILDREN ALSO, TYPE AND GEOGRAPHICAL LOCATION OF SETTLEMENT, AGE OF GP, AND VACANCY)”. THIS SENTENCE SEEMS TO HAVE COMBINED METHODS WITH AIMS/OBJECTS.
Removed some parts of this sentence to make it more objectives orientated and inserted them into Methods section.
Original:
Objectives […] controlled for other socioeconomic and GMP structural factors (GMP list size, providing care for adults only or children also, type and geographical location of settlement, age of GP, and vacancy). Methods […] GMP specific data on health reimbursement and structural indicators for secondary analysis […]
Corrected:
Objectives […] controlled for other socioeconomic and GMP structural factors. Methods […] GMP specific data on health reimbursement and structural indicators (GMP list size, providing care for adults only or children also, type and geographical location of settlement, age of GP, and vacancy) for secondary analysis […]
2
IN THE RESULTS SECTION, PLEASE FOCUS ON INTERPRETING THE MAIN FINDINGS RATHER THAN PROVIDING THE POINT ESTIMATES ONLY. FOR EXAMPLE, WHAT DOES IT MEAN TO HAVE A B=0.199. ALSO, PLEASE PROVIDE CI AND NOT P-VALUES HERE.
The result section has been modified according to these suggestions
Original:
Results: The srEDU (b=0.199, p<0.001) and srEMP (b=-0.282, p<0.001) had significant effect on reimbursements. GP age >65 (b=-0.026, p<0.001), list size <800 (b=-0.043, p<0.001) and 800-1200 (b=-0.018, p=0.010), and GMP providing adults only (b=0.199, p<0.001) also had significant association with reimbursements. There was also significant variability across counties. However, rRP proved to be a not significant factor (b=0.002, p=0.250).
Corrected:
Results: The srEDU had significant positive (b=0.199, 95%CI: 0.128;0.271) and the srEMP had significant negative (b=-0.282, 95%CI: -0.359;-0.204) effect on reimbursements. GP age >65 (b=-0.026, 95%CI: -0.036;-0.016), list size <800 (b=-0.043, 95%CI: -0.066;-0.020) and 800-1200 (b=-0.018, 95%CI: -0.031;-0.004]), had significant negative association with reimbursements, and GMP providing adults only (b=0.016, 95%CI: 0.001;0.032) had a positive one. There was also significant reimbursement variability across counties. However, rRP proved to be a not significant influencing factor (b=0.002, 95%CI: -0.001;0.005).
3.
OECD- PLEASE PROVIDE THE FULLFORM; ALSO, PLEASE TRY TO LIMIT THE ABBREVIATIONS USED.
The abbreviation of OECD has been replaced with the full name in the first sentence of the Introduction. The same has been done for GDP.
Original:
Healthcare expenditure has risen dramatically during the last decades in the OECD countries, where it accounts nowadays for 8.8% of the GDP [1]
Corrected:
Healthcare expenditure has risen dramatically during the last decades in the Organization for Economic Co-operation and Development countries, where it accounts nowadays for 8.8% of the gross domestic product [1]
4.
METHODS
THERE ARE WAY TOO MANY ACRONYMS IN THIS SECTION. PLEASE TRY AND LIMIT THE USE OF ACRONYMS.
4a
The full terms (instead abbreviations) were used for OECD and GDP in the first sentence of the introduction. (Both OECD and GDP have been removed from the List of abbreviation section)
Original:
Healthcare expenditure has risen dramatically during the last decades in the OECD countries, where it accounts nowadays for 8.8% of the GDP [1]
Corrected:
Healthcare expenditure has risen dramatically during the last decades in the Organization for Economic Co-operation and Development countries, where it accounts nowadays for 8.8% of the gross domestic product [1]
4b
The explanation for GMP in the List of abbreviation section needed modification (but, the solution for GMP was good both in the abstract and in the Introduction):
Original:
GMP: General Medical Practitioner
Corrected:
GMP: General Medical Practice
4c
The abbreviation for National Health Insurance Fund was defined two times in the main text. The second one has been deleted (in the Setting section):
Original:
obtained from the National Health Insurance Fund (NHIF).
Corrected:
obtained from the NHIF.
4d
We tried to use full terms instead of the abbreviations for explanatory variables but the text resulted was much more difficult to read. We hope that you can accept that the application of these abbreviations have not been changed.
5.
SINCE EXPENDITURES CANNOT BE LESS THAN 0, DID THE AUTHORS EXPLORE USING OTHER DISTRIBUTIONS LIKE GAMMA?
Thank you for this question! We did not face any problems during the analysis which warranted the application of gamma distribution based statistical approaches.
6.
WHAT DOES ANOVA ADD WHEN THE AUTHORS ALREADY HAVE ESTIMATES FROM ADJUSTED REGRESSION MODELS?
ANOVA was used for the descriptive analysis of the crude measures in order to demonstrate the inequalities in the investigated sample, meanwhile the multivariate regression modelling was used to analyze the associations between the standardized reimbursement and explanatory parameters To emphasize it, the second sentence in statistical analysis section has been modified.
Original:
Associations between socioeconomic characteristics of patients and structural characteristics of GMPs and srEXP were tested with the Pearson correlation and one-way ANOVA.
Corrected:
Uneven distribution of srEXP by socioeconomic characteristics of patients and structural characteristics of GMPs were tested with the Pearson correlation and one-way ANOVA.
7.
ON WHAT BASIS WERE THE COVARIATES TO BE ADJUSTED FOR CHOSEN?
As mentioned in the introduction (paragraphs 3,4,5), there are well establish connections between our covariates (socioeconomic status of patients and structural characteristics of general medical practices) and health expenditure. In order to emphasize that the study was to determine whether Roma ethnicity has distinct effect on the reimbursement additional to the effect of these well-known factors, the last sentence of the last paragraph of the Introduction has been modified as follows.
Original:
The study aimed to describe the usefulness of GMP level Roma-specific indicator in formulating inequality-reducing health policy.
Corrected:
Our investigation aimed at describing the Roma effect independent from these well descried determinants of expenditures, as well as the usefulness of general medical practice level Roma-specific indicator in formulating inequality-reducing health policy.
8.
INSTEAD OF JUST PROVIDING THE BETA AND P-VALUES, PLEASE RATHER FOCUS ON INTERPRETATION AND PROVIDE CI INSTEAD OF OR ALONG WITH P-VALUES
8a
We applied beta (standardized linear regression coefficient) to demonstrate the relative impact of the explanatory factors. According to your suggestion the Figure 2, which present the betas, have been completed with error bars representing the 95%CIs for betas.
8b
The title of the figure has been modified also.
Original:
Figure 2. Strength of the association between socioeconomic factors, including Roma population proportion, and GMP-specific structural indicators with average per-capita GMP-specific expenditures of the National Health Insurance Fund based on the standardized linear regression.
Corrected:
Figure 2. Strength of the association between socioeconomic factors, including Roma population proportion, and GMP-specific structural indicators with average per-capita GMP-specific expenditures of the National Health Insurance Fund based on the standardized linear regression coefficients from multivariate regression model.
8c
In the last paragraph of the Results section, the betas had been completed with the corresponding 95%Cis.
Original:
According to the standardized linear regression coefficients, srEMP (β=-0.219) had the strongest negative effect on spending, followed by being located in Győr-Moson-Sopron county (β=-0.140) and Veszprém county (β=-0.097). The strongest positive effect on spending came from being located in Baranya county (β=0.159) followed by srEDU (β=0.13). The nonsignificant effect of the Roma population proportion was associated with a very small standardized linear regression coefficient (β=-0.023).
Corrected:
According to the standardized linear regression coefficients, srEMP (β=-0.219, 95% CI: -0.277;-0.159) had the strongest negative effect on spending, followed by being located in Győr-Moson-Sopron county (β=-0.140, 95% CI: -0.172;-0.108) and Veszprém county (β=-0.097, 95% CI: -0.128;-0.065). The strongest positive effect on spending came from being located in Baranya county (β=0.159, 95% CI: 0.126;0.193) followed by srEDU (β=0.13, 95% CI: 0.085;0.181). The nonsignificant effect of the Roma population proportion was associated with a very small standardized linear regression coefficient (β=-0.023, 95% CI: -0.017;0.063).

Reviewer 3 Report
The article “Self-declared Roma ethnicity and health insurance reimbursement: a nationwide cross-sectional investigation at the general medical practice level in Hungary” presented a large cross-sectional analysis of health insurance spending in Hungarian populations. Please find below my detailed comments. I hope these comments can help the researchers improve the manuscript.
Introduction
Discussions in the third paragraph are inconsistent with those in the first paragraph: If health expenditure is driven by changing needs and advancement of medical technologies, why “Obviously, demographic (age and sex, type of living place[13–15]) and socio-economic status (e.g. education [16–18], employment [19–22]etc.) of clients determine the expenditures”? There is something missing in the logic.
Similarly, the discussions about high risks of the Roma community are about “health”, not “health spending”. Why?
While I agree that primary care is important to ensure health equality, I don’t understand the rationale of the discussion here. Please note that health equality is different from equality in health spending (or reimbursement). The authors also used the term GMP without an explanation. GMP is not a term self-explainable.
Methods
The authors need to describe the health financing and payment system in Hungary. Otherwise, readers would not be able to understand why there is a need to study “reimbursement”. It appears that there are out-of-pocket payments involved in GMP services. Is that the case?
What does “settlement” mean? Why were indicators (such as education and employment) standardised for each settlement? This is very confusing. Why don’t use raw data collected from each GMP?
“the resulted variables were transformed into normal distribution”: what variables? Health spending and reimbursement?
GMPs were also categorised by counties: County is a higher level of variable. Was a two-level modelling adopted?
What is NHIF? What does it mean “GMP financing by NHIF does not influence the per capita health spending variability”? Financing can not be considered equivalent to health spending.
Disease profile is perhaps the most powerful predictor of health spending. Why wasn’t it considered in the modelling?
Results
Why was average per-capita expenditure presented? Medical expenditures are not normally distributed.
Fig 2: statistical significance can be easily shown using 95% CI of Beta coefficients.
Discussion
Analyses in this study were based on GMPs: they are not based on individual patients. Please not one patients can visit a GP multiple times. Conclusions have to be cautious.
Author Response
Dear Editor and Reviewer,
Thank you very much for the careful evaluation of our manuscript “Self-declared Roma ethnicity and health insurance reimbursement: a nationwide cross-sectional investigation at the general medical practice level in Hungary”. Each comment and suggestion has been considered. Answers and responses along with the modifications we made are the following (Reviewer’s comments and questions are in capitals):
Reviewer-3:
1.
DISCUSSIONS IN THE THIRD PARAGRAPH ARE INCONSISTENT WITH THOSE IN THE FIRST PARAGRAPH: IF HEALTH EXPENDITURE IS DRIVEN BY CHANGING NEEDS AND ADVANCEMENT OF MEDICAL TECHNOLOGIES, WHY “OBVIOUSLY, DEMOGRAPHIC (AGE AND SEX, TYPE OF LIVING PLACE [13–15]) AND SOCIO-ECONOMIC STATUS (E.G. EDUCATION [16–18], EMPLOYMENT [19–22]ETC.) OF CLIENTS DETERMINE THE EXPENDITURES”? THERE IS SOMETHING MISSING IN THE LOGIC.
Thank you for this notice! We have to admit that we did not mention in the manuscript that (1) the socio-economic status of patients has profound impact on the diseases spectrum of people, consequently on the needs, and (2) this status along with the structural characteristics of primary care services influence the availability of the high quality care. Consequently, both needs and high-tech availability are related to the studied factors.
Following sentence has been inserted in to the third paragraph of the Introduction:
The socio-economic status of patients has profound impact on the diseases spectrum of people, consequently on the needs, and on the availability of the high quality care.
Following sentence has been inserted in to the fourth paragraph of the Introduction:
Further, the GP can intervene to diminish the medical high-tech availability reducing effect of the disadvantageous social status of patients.
2.
SIMILARLY, THE DISCUSSIONS ABOUT HIGH RISKS OF THE ROMA COMMUNITY ARE ABOUT “HEALTH”, NOT “HEALTH SPENDING”. WHY?
WHILE I AGREE THAT PRIMARY CARE IS IMPORTANT TO ENSURE HEALTH EQUALITY, I DON’T UNDERSTAND THE RATIONALE OF THE DISCUSSION HERE. PLEASE NOTE THAT HEALTH EQUALITY IS DIFFERENT FROM EQUALITY IN HEALTH SPENDING (OR REIMBURSEMENT).
Health and spending are not similar in this respect because the poor health of Roma is fairly described but it is not described how this poor health affects spending.
The knowledge on the financing aspects of the special health care use of Roma people is poor. (In this context we considered the financing as a summary measure of the health care use.) Because some services is underused some others are overused by Roma people, it is not trivial, whether the summary impact on financing is positive or negative.
Accordingly, the sentence in the sixth paragraph has been modified.
Original:
To conclude, the full effect of Roma population on health expenditure is still not sufficiently documented…
Corrected:
To conclude, the full effect of Roma population on the use of different services of health care and on the financing altogether is still not sufficiently documented
3.
THE AUTHORS ALSO USED THE TERM GMP WITHOUT AN EXPLANATION. GMP IS NOT A TERM SELF-EXPLAINABLE.
The definition of GMP was located not properly in the last paragraph of the introduction. This mistake has been corrected. The definition was deleted from the original place, and inserted into the fourth paragraph where the term is mentioned first.
4.
METHODS
THE AUTHORS NEED TO DESCRIBE THE HEALTH FINANCING AND PAYMENT SYSTEM IN HUNGARY. OTHERWISE, READERS WOULD NOT BE ABLE TO UNDERSTAND WHY THERE IS A NEED TO STUDY “REIMBURSEMENT”. IT APPEARS THAT THERE ARE OUT-OF-POCKET PAYMENTS INVOLVED IN GMP SERVICES. IS THAT THE CASE?
Thank you for this suggestion! The 2.4 Outcome variable section has been thoroughly restructured. It contains the requested details. (This study was on the NHIF reimbursement not on out-of-pocket payments.)
Original:
Hungarian GMPs are financed basically by capitation fees from NHIF. It is independent of services delivered by any providers from any level of healthcare. Therefore, GMP financing by NHIF does not influence the per capita health spending variability.
The total of expenditures registered by the NHIF without the per capita financing for GMPs for primary medical service provision for the 5 years of the investigation among adults were aggregated for GMPs. Expected 5-year payment were also calculated for each GMP by the 5-year per capita age and sex specific national reference payments. (Appendix Table A3) The calculated ratio of the observed and expected per capita payments resulted the GMP specific standardized relative expenditure (srEXP).
Corrected:
Health reimbursement from the NHIF is basically divided by capitation fees and performance-based fees in Hungary. GMPs are financed by capitation fees independent of services delivered for their patients by any providers from any level of healthcare. It takes only into consideration the number and demographic structure of the clients belong to the GMP. Obviously, the average of the per capita age- and sex-standardized financing for clients belong to GMP is constant across the country, and not influenced by the used health services. The outpatient and inpatient secondary care, medicine and medical device reimbursement, dental care, dialysis, imaging, transportation of patient, home nursing, and hospice are financed by performance. Therefore, the variability of the GMP level average financing by NHIF for clients belong to a GMP does generated by the use of these services. [64].
The total of expenditures registered by the NHIF for adults belong to a GMP for primary medical service provision for the 5 years of the investigation, without the per capita financing of primary medical care, were aggregated for GMPs as average per capita per year expenditure. Expected 5-year payment were also calculated for each GMP by the 5-year per capita age and sex specific national reference payments. (Appendix Table A3) The calculated ratio of the observed and expected per capita payments resulted the GMP specific standardized relative expenditure (srEXP).
5.
WHAT DOES “SETTLEMENT” MEAN? WHY WERE INDICATORS (SUCH AS EDUCATION AND EMPLOYMENT) STANDARDISED FOR EACH SETTLEMENT? THIS IS VERY CONFUSING. WHY DON’T USE RAW DATA COLLECTED FROM EACH GMP?
Subjects of our study were GMPs. The reimbursement data and the GMP structural indicators were provided by the NHIF for groups of clients belong to the same GMP. Therefore, we could use these data directly in the analysis.
Unfortunately, the socioeconomic status and Roma proportion indicators were not available for settlements. Data to compute these indicators are available only for settlements (villages and towns). Knowing by the NHIF’s database the living place of clients belong to a GMP, the settlement level indicators could be transformed into GMP level indicators.
To improve the clarity, a sentence has been inserted into the Setting section (“Subjects of this investigations were the Hungarian GMPs.”) and into the 2.2 section (“Data to compute socioeconomic indicators were provided by the Hungarian Central Statistical Office for settlements (residential places of the people).”)
6.
“THE RESULTED VARIABLES WERE TRANSFORMED INTO NORMAL DISTRIBUTION”: WHAT VARIABLES? HEALTH SPENDING AND REIMBURSEMENT?
The sentence of the last paragraph of the 2.2 section was transformed to add the missed explanation.
Original:
Next, the resulted variables were transformed into normal distribution using the two-step Box-Cox method
Corrected:
Next, the resulted variables of srEMP, srEDU, srEXP, sRP were transformed into normal distribution using the two-step Box-Cox method
7.
GMPS WERE ALSO CATEGORISED BY COUNTIES: COUNTY IS A HIGHER LEVEL OF VARIABLE. WAS A TWO-LEVEL MODELLING ADOPTED?
Yes, a mixed 2-level regression model was used in SPSS taking county clustering into account. Changes were made to the test to make that clear.
Original:
A multivariate linear regression model was used to investigate the influence of the GMP-specific socioeconomic indicators (rRP, srEDU, srEMP, rHD) and structural indicators (GP age and vacancy, type of settlement, and GMP type, size measured by number of clients, and geographical location measured by county).
Corrected:
A mixed two-level multivariate linear regression model was used to investigate the influence of the GMP-specific socioeconomic indicators (rRP, srEDU, srEMP, rHD) and structural indicators (GP age and vacancy, type of settlement, and GMP type, size measured by number of clients, and geographical location measured by county taking into account the clustering effect of the counties.
8.
WHAT IS NHIF?
The definition for the abbreviation is located in the last paragraph of the Introduction where it is mentioned first.
Some explanation about the role of NHIF relevant for our study has been added to the Setting section.
Original:
GMP-specific data on adult health expenses for the period 2012–2016 and the structural characteristics of GMPs for 2012 were obtained from the National Health Insurance Fund (NHIF).
Corrected:
GMP-specific data on adult health expenses for the period 2012–2016 and the structural characteristics of GMPs for 2012 were obtained from the NHIF which is contracted with each GMP operating in Hungary.
9.
WHAT DOES IT MEAN “GMP FINANCING BY NHIF DOES NOT INFLUENCE THE PER CAPITA HEALTH SPENDING VARIABILITY”? FINANCING CAN NOT BE CONSIDERED EQUIVALENT TO HEALTH SPENDING.
The 2.4 Outcome variable section has been restructured. See above by the Comment-5.
10.
DISEASE PROFILE IS PERHAPS THE MOST POWERFUL PREDICTOR OF HEALTH SPENDING. WHY WASN’T IT CONSIDERED IN THE MODELLING?
We agree completely and it is a limitation of our study. A new paragraph has been inserted into the Strengths and limitations section:
“An important limitation of this study was that disease profile, as a direct and important determinant of the health care use, was not among the controlled confounding factors. But, the confounding effect of disease profile was partly controlled, since adjusting factors such as age, sex, education, employment, and urbanization are strongly associated with diseases occurrence. Therefore a significant proportion of the confounding effect of disease profile was controlled by using the listed factors either in standardization or in multivariate regression modelling. Obviously, further studies are needed to determine the importance of the confounding effect of the disease pattern.”
11.
RESULTS
WHY WAS AVERAGE PER-CAPITA EXPENDITURE PRESENTED? MEDICAL EXPENDITURES ARE NOT NORMALLY DISTRIBUTED.
The average expenditure for clients belong to a GMP is the primary outcome data for our analysis, and the wide variability of this demonstrates that the inequality is great and the understanding the factors behind this inequality is important.
The sentence of the first paragraph of Results has been completed accordingly (and the fitted normal distribution curve was added to the Figure 1).
Original:
The average per-capita yearly GMP-specific expenditures were normally distributed (Figure 1).
Corrected:
The average per-capita yearly GMP-specific expenditures showed also wide variability and were normally distributed (Figure 1).
12.
FIG 2: STATISTICAL SIGNIFICANCE CAN BE EASILY SHOWN USING 95% CI OF BETA COEFFICIENTS.
Thank you for the suggestion, 95%CIs have been added to the Figure 2.
13.
DISCUSSION
ANALYSES IN THIS STUDY WERE BASED ON GMPS: THEY ARE NOT BASED ON INDIVIDUAL PATIENTS. PLEASE NOT ONE PATIENTS CAN VISIT A GP MULTIPLE TIMES. CONCLUSIONS HAVE TO BE CAUTIOUS.
Thanks for this comment! It is true, that our study used GMP level aggregated data, each conclusions are valid for this level; and we did not declare it explicitly in the conclusion. To add this declaration, the second sentence of the Conclusions section has been modified.
Original:
Our results showed very large disparities and inequality in healthcare spending across Hungarian counties.
Corrected:
Our GMP level aggregated data based investigation showed very large disparities and inequality in healthcare spending across Hungarian counties.

Round 2
Reviewer 1 Report
The amendments are done. Although, do not forget to change the unreadable figures.
Author Response
Dear Editor and Reviewer,
Thank you very much again for the useful comments and suggestions to our manuscript “Self-declared Roma ethnicity and health insurance reimbursement: a nationwide cross-sectional investigation at the general medical practice level in Hungary”. No comments received from Reviewer-2.
In addition to enhancing the overall English of the text (the modifications are shown in the text by tracked changes), answers and responses along with the modifications we made are the following (comments are in capitals):
Reviewer-1:
THE AMENDMENTS ARE DONE. ALTHOUGH DO NOT FORGET TO CHANGE THE UNREADABLE FIGURES.
We are glad that our amendments were satisfactory. Enhanced figures have been inserted into the text and uploaded as attached files also.

Reviewer 3 Report
I can see the authors made efforts to address my concerns. But I still have some major concerns.
I am very confused with the authors’ use of “health spending, health expenditure, reimbursement, payments”. They mean different things; but were used in the manuscript interchangeably without clear clarifications. Would higher spending in primary care leads to lower total health expenditure? I guess “reimbursement” indicates funds paid to GMPs by the NHIF. But it can also mean funds paid to the patients by the NHIF.
The revised manuscript did not describe the insurance system clearly. If variability of payments to GMPs depend on performance, how was performance assessed?
The study “investigated the relation between the proportion of self-declared Roma people in the population served by GMPs and payments to health services by the National Health Insurance Fund (NHIF) aggregated to the GMP level”. What was the hypothesis: higher or lower payments were expected for the GMPs serving higher proportions of Roma people? A fundamental flaw is that we don’t even know whether higher proportions of Roma people at the “settlement” level would translate into higher proportions of visits of Roma people to GMPs. Are there “reimbursement” data for Roma people?
Although sociodemographic characteristics can predict health expenditures, they can not replace health and disease variables.
Author Response
Dear Editor and Reviewer,
Thank you very much again for the useful comments and suggestions to our manuscript “Self-declared Roma ethnicity and health insurance reimbursement: a nationwide cross-sectional investigation at the general medical practice level in Hungary”. No comments received from Reviewer-1.
In addition to enhancing the overall English of the text (the modifications are shown in the text by tracked changes), answers and responses along with the modifications we made are the following (comments are in capitals):
Reviewer 3:
1.
I CAN SEE THE AUTHORS MADE EFFORTS TO ADDRESS MY CONCERNS. BUT I STILL HAVE SOME MAJOR CONCERNS.
We were happy to address your comments and concerns. It enriched our paper with your valuable opinions and experience.
2.
I AM VERY CONFUSED WITH THE AUTHORS’ USE OF “HEALTH SPENDING, HEALTH EXPENDITURE, REIMBURSEMENT, PAYMENTS”. THEY MEAN DIFFERENT THINGS; BUT WERE USED IN THE MANUSCRIPT INTERCHANGEABLY WITHOUT CLEAR CLARIFICATIONS. WOULD HIGHER SPENDING IN PRIMARY CARE LEADS TO LOWER TOTAL HEALTH EXPENDITURE? I GUESS “REIMBURSEMENT” INDICATES FUNDS PAID TO GMPS BY THE NHIF. BUT IT CAN ALSO MEAN FUNDS PAID TO THE PATIENTS BY THE NHIF.
National Health Insurance Fund pays for general medical practices by the number of clients belong to the general medical practice not taking into account the services they actually use (per capita financing).
National Health Insurance Fund pays for actually used services in the outpatient and inpatient secondary care, dental care, dialysis, imaging, transportation of patient, home nursing, and hospice (performance based reimbursement).
The cost of medications and medical devices are shared between the National Health Insurance Fund and patients themselves (co-payment).
Since per capita financing for general medical practices are not variable component of health care financing, our study focused on the other two groups of the National Health Insurance Fund payment, which is mentioned in the text as the non per capita expenditures of the National Health Insurance Fund.
- A) Modifications were made to the text to better explain this.
Original
2.4. Outcome Variable
Health payment from the NHIF is basically divided between capitation fees and performance-based fees in Hungary. GMPs are financed by capitation fees independent from services delivered by any providers from any level of healthcare. It takes only into consideration the number and demographic structure of the clients belong to the GMP. Obviously, the average of the per capita age- and sex-standardized financing for clients belong to GMP is constant across the country, and not influenced by the used health services.
The outpatient and inpatient secondary care, dental care, dialysis, imaging, transportation of patient, home nursing, and hospice are financed by performance, reimbursed by provided services. Medicines and medical devices however are financed by a co-payment system between the NHIF and patients. Therefore, the variability of the GMP level average financing by NHIF for clients belong to a GMP is generated by these two types of financing. [64].
The total of performance based reimbursements and co-payments for medicines and medical devices registered by the NHIF for adults belong to a GMP for primary medical service provision for the 5 years of the investigation, without the per capita financing of primary medical care, were aggregated for GMPs. Expected 5-year expenditures were also calculated for each GMP by the 5-year per capita age and sex specific national reference payments. (Appendix Table A3) The calculated ratio of the observed and expected expenditures resulted the GMP specific standardized relative expenditure (srEXP).
Corrected
2.4. Outcome Variable
Health payment from the NHIF is basically divided between capitation fees and performance-based fees in Hungary. GMPs are financed by capitation fees independent from services delivered by any providers from any level of healthcare. It takes only into consideration the number and demographic structure of the clients belong to the GMP. Obviously, the average of the per capita age- and sex-standardized financing for clients belong to GMP is constant across the country, and not influenced by the used health services.
The outpatient and inpatient secondary care, dental care, dialysis, imaging, transportation of patient, home nursing, and hospice are financed by performance, reimbursed by provided services. Medicines and medical devices however are financed by a co-payment system between the NHIF and patients. Therefore, the variability of the GMP level average financing by NHIF for clients belong to a GMP is generated by these two types of financing. [64].
The total of performance based reimbursements and co-payments for medicines and medical devices registered by the NHIF for adults belong to a GMP for primary medical service provision for the 5 years of the investigation, without the per capita financing of primary medical care, were aggregated for GMPs. Expected 5-year expenditures were also calculated for each GMP by the 5-year per capita age and sex specific national reference payments. (Appendix Table A3) The calculated ratio of the observed and expected expenditures resulted the GMP specific standardized relative expenditure (srEXP).
- B) The title of the manuscript has been corrected as well:
Original
Self-declared Roma ethnicity and health insurance reimbursement: a nationwide cross-sectional investigation at the general medical practice level in Hungary
Corrected
Self-declared Roma ethnicity and health insurance expenditures: a nationwide cross-sectional investigation at the general medical practice level in Hungary
- C) The term of “reimbursement” has been replaced with expenditure or the abbreviation of srEXP in the abstract:
Original
average per capita health care reimbursement
[…] GMP specific data on health reimbursement and structural indicators
[…] applied to evaluate the relationship between socioeconomic and structural indicators with reimbursements.
[…] the srEMP had significant negative (b=-0.282, 95%CI: -0.359;-0.204) effect on reimbursements.
[…] had significant negative association with reimbursements,
[…] There was also significant reimbursement expenditure variability across counties.
[…] small practice size was associated with lower spending in Hungary.
[…] did not significantly affect the health reimbursement according to our GMP level study
Corrected
average per capita expenditure of the National Health Insurance Fund (NHIF)
[…] GMP specific data on health expenditures and structural indicators
[…] applied to evaluate the relationship between socioeconomic and structural indicators with srEXP.
[…] the srEMP had significant negative (b=-0.282, 95%CI: -0.359;-0.204) effect on srEXP.
[…] had significant negative association with srEXP,
[…] There was also significant expenditure variability across counties.
[…] small practice size was associated with lower NHIF expenditures in Hungary.
[…] did not significantly affect the health expenditures according to our GMP level study
- D) The setting was modified accordingly:
Original
GMP-specific data on adult health expenses
Corrected
GMP-specific data on adult health expenditures
- E) The sentence on descriptive statistics in Statistical analysis section was also modified:
Original
Expenditures for GMP categories were described by means (±SD) of srEXP.
Corrected
Per capita expenditures in a year for GMP categories as the total of performance based reimbursements and co-payments for medicines and medical devices divided by the person-years of clients were described by means (±SD).
- F) The sentence in the Descriptive statistics section of Results section was also modified:
Original
Per-capita expenses across demographic strata showed
Corrected
Per-capita expenditures across demographic strata showed
- G) Sentences in Main findings section (third paragraph) were also modified:
Original
A positive association between a patient’s educational attainment and insurance spending on health was observed in our investigation.
[…] ultimately leading to higher usage of medical services thus increasing spending.
[…] these two variables were inversely related to spending.
Corrected
A positive association between a patient’s educational attainment and NHIF expenditures was observed in our investigation.
[…] ultimately leading to higher usage of medical services thus increasing NHIF expenditures.
[…] these two variables were inversely related to NHIF expenditures.
- H) Sentence in Main findings section (third paragraph) were also modified:
Original
Geographical location was also found to have strong impact on reimbursements.
Corrected
Geographical location was also found to have strong impact on NHIF expenditures.
- I) Sentence in Strengths and limitations section (5th paragraph) was also modified:
Original
Our results are not about the determinants of the Roma individuals’ NHIF health reimbursement, but about the role of the proportion of Roma in a GMP on the average per capita per year NHIF reimbursement.
Corrected
Our results are not about the determinants of the Roma individuals’ NHIF expenditures, but about the role of the proportion of Roma in a GMP on the average per capita per year NHIF expenditure.
- J) Sentences in Conclusions section were also modified:
Original
showed very large disparities and inequality in healthcare spending across Hungarian counties.
[…] The self-reported Roma population had a nonsignificant association with health spending;
Corrected
showed very large inequality in NHIF expenditures across Hungarian counties.
[…] The self-reported Roma population had a nonsignificant association with NHIF expenditures;
3.
THE REVISED MANUSCRIPT DID NOT DESCRIBE THE INSURANCE SYSTEM CLEARLY. IF VARIABILITY OF PAYMENTS TO GMPS DEPENDS ON PERFORMANCE, HOW WAS PERFORMANCE ASSESSED?
The section of the Outcome variable has been modified, which the modified text declares the variable and invariable parts of the National Health Insurance Fund payments.
4.
THE STUDY “INVESTIGATED THE RELATION BETWEEN THE PROPORTION OF SELF-DECLARED ROMA PEOPLE IN THE POPULATION SERVED BY GMPS AND PAYMENTS TO HEALTH SERVICES BY THE NATIONAL HEALTH INSURANCE FUND (NHIF) AGGREGATED TO THE GMP LEVEL”. WHAT WAS THE HYPOTHESIS: HIGHER OR LOWER PAYMENTS WERE EXPECTED FOR THE GMPS SERVING HIGHER PROPORTIONS OF ROMA PEOPLE?
Our study is a descriptive one, we wanted to produce the very first results on the relation between Roma ethnicity and health expenditures as stated in the objectives. However, Roma’s known low health status compared to the reference population should in theory correlate with more use of health services; meaning higher NHIF average per capita spending to GMPs serving higher proportions of Roma. On the other hand, Roma populations might not adequately use health services demanded by their lower health status, either due to discrimination or other reasons which will be reflected by lower NHIF average per capita spending, especially if this discrimination counterbalances the higher need for services.
Sentence added to the Objectives section of the introduction that reflects that thought:
As well as to investigate whether proportion of Roma is positively associated with the NHIF expenditures reflecting that the poor health of Roma meets adequate care, or this correlation is negative demonstrating that Roma's poor access to health care counterbalances their increased needs.
5.
A FUNDAMENTAL FLAW IS THAT WE DON’T EVEN KNOW WHETHER HIGHER PROPORTIONS OF ROMA PEOPLE AT THE “SETTLEMENT” LEVEL WOULD TRANSLATE INTO HIGHER PROPORTIONS OF VISITS OF ROMA PEOPLE TO GMPS. ARE THERE “REIMBURSEMENT” DATA FOR ROMA PEOPLE?
The payment that was investigated in this study was absolutely independent of how many times Roma visited a GP. GPs are financed by not performance based manner. A client not visiting GP and another one visiting GP many times are financed in the same way.
There are no direct reimbursement data for Roma specifically, since the NHIF does not register ethnicity at all.
6.
ALTHOUGH SOCIODEMOGRAPHIC CHARACTERISTICS CAN PREDICT HEALTH EXPENDITURES, THEY CAN NOT REPLACE HEALTH AND DISEASE VARIABLES.
The conclusions had been restructured accordingly:
Original
Equality in treatment and health care access are fundamental concepts in healthcare systems, especially in Europe, but monitoring the effective application of this goal is a major challenge.
Our GMP level aggregated data based investigation showed very large disparities and inequality in healthcare spending across Hungarian counties. Moreover, higher education and employment had significant effects on health, increasing it and decreasing it, respectively, thus making them prime targets for intervention. The self-reported Roma population had a nonsignificant association with health spending; however, these findings do not necessarily hold for other groups of Roma, such as segregated and traveling Roma. Further studies and data collection are recommended in order to more accurately measure the effect of the Roma population on health expenditure. There was no significant relation between density and health expenditure; however, further studies are needed to investigate this result.
Corrected
Our GMP level aggregated data based investigation showed very large inequality in NHIF expenditures across Hungarian counties. The higher level of education and unemployment had significant positive effect on NHIF expenditure, making them prime targets for intervention. The self-reported Roma population had a nonsignificant association with NHIF expenditures. These ecological observations do not suggest that Roma’s access to the health care is restricted. These observations suggest that the poor access of Roma to health care is weaker or not effective risk factor of the reduced NHIF financing than their poor health status determined increased needs.